# Lipoprotein (a), Inflammation, and Atherosclerosis

**DOI:** 10.3390/jcm12072529

**Published:** 2023-03-27

**Authors:** Stefania Angela Di Fusco, Aldo Pietro Maggioni, Pietro Scicchitano, Marco Zuin, Emilia D’Elia, Furio Colivicchi

**Affiliations:** 1Clinical and Rehabilitation Unit, San Filippo Neri Hospital, ASL Rome 1, 00135 Rome, Italy; 2ANMCO Research Center, Heart Care Foundation, 50121 Florence, Italy; 3Cardiology Department, Hospital “F. Perinei” ASL, 70022 Altamura, Italy; 4Department of Translational Medicine, University of Ferrara, 44121 Ferrara, Italy; 5Cardiology Unit, Cardiovascular Department, ASST Papa Giovanni XXIII, 24127 Bergamo, Italy

**Keywords:** lipoprotein (a), cardiovascular prevention, inflammation, colchicine, personalized medicine

## Abstract

Growing evidence has shown that high levels of lipoprotein (a) (Lp(a)) and chronic inflammation may be responsible for the residual risk of cardiovascular events in patients managed with an optimal evidence-based approach. Clinical studies have demonstrated a correlation between higher Lp(a) levels and several atherosclerotic diseases including ischemic heart disease, stroke, and degenerative calcific aortic stenosis. The threshold value of Lp(a) serum concentrations associated with a significantly increased cardiovascular risk is >125 nmol/L (50 mg/dL). Current available lipid-lowering drugs have modest-to-no impact on Lp(a) levels. Chronic inflammation is a further condition potentially implicated in residual cardiovascular risk. Consistent evidence has shown an increased risk of cardiovascular events in patients with high sensitivity C reactive protein (>2 mg/dL), an inflammation biomarker. A number of anti-inflammatory drugs have been investigated in patients with or at risk of cardiovascular disease. Of these, canakinumab and colchicine have been found to be associated with cardiovascular risk reduction. Ongoing research aimed at improving risk stratification on the basis of Lp(a) and vessel inflammation assessment may help refine patient management. Furthermore, the identification of these conditions as cardiovascular risk factors has led to increased investigation into diagnostic and therapeutic strategies targeting them in order to reduce atherosclerotic cardiovascular disease burden.

## 1. Introduction

Despite improved treatments that target traditional cardiovascular risk factors with increasing efficacy, atherosclerotic diseases remain responsible for a substantial number of adverse events, thus basic and clinical research has focused on identifying and managing additional risk factors that impact the residual risk. Among these, elevated circulating levels of lipoprotein (a) (Lp(a)) and chronic subclinical inflammation seem to play a non-negligible role in atherosclerotic disease. Furthermore, Lp(a) pathophysiologic effects and inflammatory processes share common biological pathways that contribute to atherogenesis. Identifying these conditions as atherosclerotic risk factors and developing interventions aimed at managing these conditions may help to reduce atherosclerotic cardiovascular disease (CVD) burden. In this review, we aim to summarize updated evidence supporting the role of high Lp(a) levels and inflammation as atherosclerotic disease risk factors and report diagnostic approaches available or under investigation to identify these conditions in clinical practice. Furthermore, we also discuss possible therapeutic interventions to reduce the adverse impact of these conditions.

## 2. Lipoprotein (a) as an Atherogenic Factor

### 2.1. Clinical Evidence and Therapeutic Approach

Lp(a) is a low-density lipoprotein (LDL)–like particle that is composed of triglycerides, cholesteryl esters, oxidized phospholipids, and a molecule of apolipoprotein B-100 (apoB) bound to apolipoprotein (a) (apo(a)). Lp(a) has been associated with pro-inflammatory, pro-atherosclerotic, and pro-thrombotic effects, which may promote the development and progression of several cardiovascular diseases regardless of the presence of traditional cardiovascular risk factors (Figure 1) [1].

Lp(a) particles can cross the endothelial barrier, be retained in the arterial wall, and promote atherosclerotic plaque growth. Oxidized phospholipids carried by Lp(a) trigger macrophage apoptosis and may promote atherosclerotic lesion transformation into “instable” plaques. Lp(a) seems to contribute to arterial vessel wall inflammation by promoting monocyte cell extravasation and endothelial cell activation [2]. Experimental studies have shown that these effects may be ascribed to adhesion molecules, e.g., intercellular adhesion molecule-1 (ICAM-1), transcription and translation upregulation and to increased activity of the enzyme 6-phophofructo-2-kinase/fructose-2,6-biphosphatase (PFKFB)-3 induced by Lp(a) [2].

In addition, apo(a) KIV domains seem to be involved in the interaction with beta2-integrin Mac-1, which induces nuclear factor kB (NFkB) activation and leads to the production of molecules that mediate the adhesion of monocytes to the endothelium and subsequent arterial wall invasion [3].

In vitro studies have also found that apo(a) is able to stimulate vascular smooth muscle cell proliferation and migration [4]. The apo(a) KIV10 domain seems to interact with plasminogen receptors on the cell surface and in the extracellular matrix, thus competing in fibrinolytic processes [5]. The binding of Lp(a) to fibrin prevents plasminogen activation and results in impaired clot degradation [6]. Furthermore, Lp(a) has been found to be able to bind and inactivate tissue factor pathway inhibitors [7]. Of note, most of the mechanisms that explain potential Lp(a) prothrombotic effects have been found in in vitro studies and their impact on atherothrombotic events must be confirmed in clinical settings.

Data derived from the Multi-Ethnic Study of Atherosclerosis (MESA) [8] recently outlined a 60% increase in risk for coronary heart disease (CHD) in patients with higher levels of Lp(a) (>50 mg/dL), regardless of serum concentrations of low-density lipoprotein cholesterol (LDL-C). Furthermore, when LDL-C is higher than 100 mg/dL, a 19% increase in all-cause mortality has been reported [9]. Verbeek et al. [10] observed attenuation in cardiovascular risk in patients with high Lp(a) levels and lower LDL-C values (<2.5 mmol/L) in the primary prevention setting, suggesting that intensive therapeutic approaches aimed at reducing LDL-C help to reduce cardiovascular risk even in patients with higher Lp(a).

Patients with premature acute coronary syndrome (ACS) were reported to have higher Lp(a) levels than older individuals with ACS [11]. The impact of Lp(a) levels on complications of interventional procedures to treat coronary artery disease (CAD) has been investigated in a meta=analysis including nine cohort studies. Despite heterogeneity among studies, higher levels in Lp(a) have been found to be associated with the occurrence of in-stent restenosis, with a stronger association in the Asian population [12].

Lp(a) seems to also be involved in heart valve and great artery calcification. It has been observed that high Lp(a) levels were associated with an 82%, 37%, and 36% increased risk in aortic valve, mitral valve, and thoracic aortic vessel calcification, respectively [13]. Wang et al. [14] found a 5% and 3% increase in aortic aneurysm and large artery atherosclerosis stroke incidence, respectively, in patients with higher Lp(a) levels.

The persistent risk of cardiovascular events despite lipid-lowering treatments that efficaciously reduce LDL-C serum levels supports the need of additional therapeutic targets to pursue [15]. Due to the evidence of a correlation between Lp(a) and CVD risk, treatments capable of reducing Lp(a) levels are under investigation. Currently available lipid-lowering strategies have a different impact on Lp(a) levels (Figure 2). Clinical studies have documented conflicting results on Lp(a) level changes in statin-treated patients, with no effect or a slight increase in Lp(a) being reported [16,17]. However, the clinical relevance of a statin-associated Lp(a) increase is still debated [18] and further studies are needed to establish the impact of statins on cardiovascular risk associated with Lp(a) levels. Bempedoic acid—a novel pharmacological agent that inhibits cholesterol biosynthesis—does not affect Lp(a) levels [19]. Similarly, the use of ezetimibe does not impact Lp(a) levels [20].

Graphical representation of maximum change percentage in Lp(a) serum levels with available or in late-stage clinical development lipid-lowering drugs.

Nicotin acid treatment reduces Lp(a) concentrations by >20% as reported in the meta-analysis by Sahebkar et al. [21]. The Lp(a) reduction was irrespective of the drug dosage (< or >2000 mg/day), but the drug’s adverse effects—skin flushing combined with dizziness, itching, nausea and vomiting–limit treatment compliance. Research on lipid-lowering treatments has led to the use of proprotein convertase subtilisin/kexin type 9 (PCSK9) inhibitors, which interfere with LDL-C receptor degradation promoted by the binding of PCSK9 with the receptor. Monoclonal antibodies against PCSK9 are also able to reduce Lp(a) levels up to >27% [22]. Inclisiran—a novel PCSK9 inhibitor that acts by silencing gene transcription–led to a mean reduction in Lp(a) of between 14% and 22% in phase III clinical trials [23,24].

More recently, gene silencing approaches targeting apolipoprotein(a) synthesis have been developed. SLN360, a small interfering RNA (siRNA) that inhibits apo(a) mRNA translation, reduces Lp(a) levels from 30 to 98% in relation to drug dosage [25]. Olpasiran, a further siRNA targeting apo(a) mRNA, has been found to result in an up to 100% placebo-adjusted mean percentage reduction in Lp(a) with a 225 mg dose administered every 24 weeks [26].

Pelacarsen, an antisense oligonucleotide (ASO) that inhibits apo(a) mRNA translation in hepatocytes, decreases Lp(a) by 36–80% based on drug dose [27]. Specifically, in patients with established cardiovascular disease, pelacarsen reduced Lp(a) by 80% when administered at a weekly dose of 20 mg, with a persistent effect at 113 days from last administration [27]. Furthermore, in patients with high Lp(a) serum concentrations, pelacarsen has been found to lead to lower inflammatory activity and transendothelial migration of circulating monocytes [28]. Further studies are necessary to establish the clinical impact of these effects, which were not found with powerful LDL-C lowering agents. The main adverse effects of pelacarsen seem to be limited to injection-site reactions, which are generally mild. The ongoing Lp(a)-HORIZON study (NCT04023552), which enrolled 8324 patients with established cardiovascular disease, will provide data on the effect of 80 mg of monthly subcutaneous injection of pelacarsen on cardiovascular outcomes.

Finally, Lp(a) apheresis, an invasive approach, has a pronounced effect in serum Lp(a) concentrations lowering. It has been found to reduce Lp(a) by ~70% with each single treatment [29], therefore resulting in a more consistent reduction in cardiovascular risk correlated with high Lp(a) levels than currently available pharmacological lipid-lowering treatments.

Overall, among drugs currently available in clinical practice to reduce atherosclerotic risk associated with dyslipidemia, the only agents that seem to have a possible clinically relevant impact in reducing Lp(a) are represented by PCSK9 inhibitors. Indeed, it has been estimated that to achieve a clinical benefit in terms of atherosclerotic cardiovascular risk reduction (22%) similar to that associated with an LDL-C reduction of 1 mmol/L, an Lp(a) reduction of ~240 nmol/L is necessary in primary prevention [30], while a less prominent reduction in Lp(a) levels (116 nmol/L) [31] is needed to have the same decrease in cardiovascular risk in the secondary prevention setting.

Further research is needed in order to identify the best approach to lower Lp(a) and to establish the impact of such strategies on clinical outcomes.

### 2.2. How to Measure and Quantify the Risk Associated with High Lp(a) Levels

Two main challenges limit Lp(a) level assessment in clinical practice: the absence of standardized measurement methods and established target levels. Although clinical studies have shown that atherosclerotic cardiovascular risk associated with Lp(a) rises with increasing Lp(a) concentrations, reference thresholds to identify individuals at higher risk may be useful in clinical practice [32]. In the general population, Lp(a) levels vary with ethnicity and most people have relatively low Lp(a) levels. A significant increase (>20%) in cardiovascular risk has been observed in the third of the population with the highest Lp(a) levels [32]. Overall, Lp(a) levels >250 nmol/L are associated with an increase in cardiovascular risk of around 50%. 

Since, Lp(a) levels are mostly (>90%) genetically determined [33], stable throughout life, and largely unaffected by environmental factors, except in specific clinical settings such as pregnancy and inflammatory disease, it is suggested to measure them once during adulthood as part of a lipid profile [32]. Lp(a) measurements are also recommended in youth that have experienced an ischemic stroke or who have a family history of premature CVD or high Lp(a) [32]. Cascade testing for Lp(a) assessment is recommended in relatives of individuals with familial hypercholesterolemia [32].

At present, no specific method for Lp(a) concentration measurement has been established as a gold standard. Interindividual apo(a)-size heterogeneity poses some difficulties in Lp(a) concentration quantification. Indeed, most of the available analytical methods for Lp(a) measurement use immunoassays which are apo(a)-isoform-sensitive, and therefore may overestimate or underestimate Lp(a) levels based on apo(a) size. The Marcovina assay, which uses monoclonal antibodies, is the least apo(a)-size-sensitive immunoassay currently available on the market [34]. Although some available assays measure Lp(a) in mass and others in molar units, in clinical practice it is recommended to measure Lp(a) serum concentrations in molar units, as this measurement quantifies Lp(a) particles irrespective of the particle molecular mass. Furthermore, the conversion between the two units of measurement is not advisable [1]. Recently, the liquid chromatography tandem mass spectrometry assay, an isoform-independent assay, has been validated and proposed as a reference method [35].According to the European Atherosclerosis Society consensus statement, the reference threshold value to define Lp(a) concentration as elevated, which confers a significantly increased cardiovascular risk, is >125 nmol/L (50 mg/dL), while an increased risk due to Lp(a) may be ruled out if its concentration is <75 nmol/L (<30 mg/dL) [32]. Since currently available lipid-lowering agents have a moderate effect, at best, on circulating Lp(a) levels, patients with elevated Lp(a) levels should be managed with more intensive interventions to reduce the impact of other modifiable cardiovascular risk factors [1].

## 3. Inflammation and Atherosclerotic Cardiovascular Disease

### 3.1. Clinical Evidence and Therapeutic Approach

Growing evidence has shown that systemic chronic inflammation may be a relevant promotor of atherogenesis. Although the presence of inflammatory cells in atherosclerotic lesions was first reported in the late 1800s, the role of inflammation in atherosclerosis initiation and progression has been recognized only within the last decades [36]. Indeed, inflammatory response is involved in atherosclerotic lesion development from endothelial dysfunction to plaque erosion/rupture and thrombosis.

In the field of CVD, residual inflammatory risk (RIR) represents an important but often underestimated issue. From a clinical point of view, RIR could be defined as the residual risk of incident vascular events or progression of established vascular injuries in patients treated according to the current evidence-based recommended care. Real word data show that RIR is a common finding in patients with or at risk of cardiovascular disease. In healthy individuals, the level of circulating inflammatory indexes has been found to increase proportionally with the presence of traditional atherosclerotic risk factors [37]. Conversely arterial hypertension, diabetes, dyslipidemia, weight gain, smoking, older age, and sedentary lifestyle are all inducers of inflammation and drive endothelial dysfunction [38] and other physiopathologic processes implicated in atherosclerosis. A large cohort study including apparently healthy females has shown that the risk of cardiovascular events increases with an increase in levels of high-sensitivity C-reactive protein (hsCRP), a circulating inflammatory biomarker [39]. In the Variation in Recovery: Role of Gender on Outcomes of Young AMI Patients registry, almost 50% of young post-myocardial infarction patients (≤55 years of age) had RIR (hsCRP > 3 mg/L) [40]. A large body of evidence on inflammation as a cardiovascular risk factor derives from studies that have tested lipid-lowering treatments. In the Pravastatin or Atorvastatin Evaluation and Infection Therapy (PROVE IT) [41] and the Improved Reduction of Outcomes: Vytorin Efficacy International (IMPROVE-IT) [42] trials, ~30% of patients that achieved C-LDL levels <70 mg/dL had hsCRP > 2 mg/L. Overall in several studies investigating lipid-lowering drugs, serum levels of hsCRP > 2 mg/L have been consistently proven to be associated with residual cardiovascular risk [41,42,43]. Furthermore, hsCRP has been found to predict cardiovascular risk even when very low C-LDL levels were achieved with PCSK9 inhibitors [44].

Considering the role of chronic inflammation in atherosclerotic cardiovascular disease, interventions aimed at attenuating inflammatory response have been evaluated to reduce CVD risk. Changes in lifestyle behaviors, such as stopping smoking, increasing physical activity, and reducing body weight, are associated with C-reactive protein (CRP) reduction [45] and the reduced systemic inflammation may contribute to the benefit of these behaviors in CVD prevention. A number of anti-inflammatory drugs have also been investigated in clinical studies to assess their impact on cardiovascular risk. Of note, statin treatment has been found to be associated with a reduction in cardiovascular events even in patients with normal LDL-C values if they had high CRP [46]. Randomized clinical studies that have demonstrated the cardiovascular benefits of several anti-inflammatory strategies are proof-of-concept that inflammation plays a crucial role in atherosclerotic disease. The Canakinumab Anti-inflammatory Thrombosis Outcome Study (CANTOS) [47] tested the impact of canakinumab, a human anti-interleukin (IL)-1β monoclonal antibody, in patients who had a myocardial infarction at least one month before the enrollment and with hsCRP ≥ 2 mg/dL and who were receiving standard treatment in accordance with guideline recommendations. Canakinumab 150 mg every 3 months was found to reduce major cardiovascular event (myocardial infarction, stroke or cardiovascular death) incidence by 15% compared with placebo [47], which was driven by the reduction in myocardial infarction (−24%) recurrence. The clinical use of canakinumab is limited due to the increased risk of fatal infection observed in the CANTOS study and the high cost of the drug.

The Cardiovascular Inflammation Reduction Trial (CIRT) [48] tested methotrexate, an antimetabolite that acts as an immunosuppressant, in patients with a history of or at high risk of cardiovascular events. This study was stopped because of futility. CIRT enrolled patients regardless of hsCRP levels [48]. However, the results of the study showed that methotrexate did not modify the circulating levels of inflammatory markers. Colchicine, an anti-inflammatory and anti-proliferative agent used in the cardiovascular field to treat pericarditis, has been firstly tested in the low-dose colchicine (LoDoCo) study, an open label study that showed a reduction in major cardiovascular events in patients with stable coronary artery disease [49]. Subsequently colchicine has been studied in the Colchicine Cardiovascular Outcomes Trial (COLCOT) [50]. In this study, which enrolled approximately 5000 patients who had had an ACS in the 30 days prior to enrollment, after a mean follow-up of 2.3 years, colchicine 0.5 mg/die compared to placebo reduced the primary endpoint, a composite of cardiovascular death, resuscitated cardiac arrest, myocardial infarction, stroke, and urgent hospitalization due to angina requiring coronary revascularization (hazard ratio, (HR) 0.77; 95% confidence interval (CI), 0.61–0.96; *p* = 0.02) [50]. Colchicine was also investigated in the LoDoCo2 study that enrolled patients with a history of coronary events or revascularization in the previous 6 months [51]. In this study, colchicine was also found to reduce the primary endpoint, a composite of cardiovascular death, ACS, stroke, or ischemia-driven coronary revascularization (HR, 0.69; 95% CI, 0.57–0.83; *p* < 0.001) [51]. In both studies, (COLCOT and LoDoCo2) patients were enrolled regardless of hsCRP levels.

Potential benefits of recombinant antibody against interleukin-1 (IL-1) receptor have also been investigated. Clinical studies on patients with ACS showed reduction in hsCRP plasmatic concentrations without clear benefit on ischemic risk [52]. However, treatment with anakinra, a recombinant human IL-1 receptor antagonist, has been found to be associated with a reduced risk of death and heart failure in patients with ST-segment–elevation myocardial infarction [53].

Interleukin 6 (IL-6) is a further therapeutic target of anti-inflammatory strategies. Ziltivekimab, a monoclonal antibody against IL-6, has been tested in patients with chronic kidney disease in a phase 2 trial [54]. The study showed a significant reduction in hsCRP levels, with a reduction that was almost twice that achieved with canakinumab and a good safety profile. Of note, in this study ziltivekimab also led to a dose-dependent reduction in Lp(a) levels. Tocilizumab, a monoclonal antibody against the IL-6 receptor that has been tested in patients with acute myocardial infarction, was found to reduce hsCRP and myocardial injury, as measured by troponin levels, in patients without ST-segment elevation [55] and to increase myocardial salvage, as measured by magnetic resonance, in patients with ST-segment elevation [56]. Overall, although targeting RIR seems to impact CVD burden (Table 1), further basic and clinical investigations are needed to identify anti-inflammatory therapeutic strategies, which, when added to current evidence-based optimal treatment, can reduce cardiovascular events with a good safety profile. Overall, results of studies that focus on colchicine in CVD are encouraging [57]. Indeed, this is a low-cost drug already widely available in clinical practice. Clinical studies have shown that prognosis of patients with CVD and treated with established secondary prevention therapies may be further improved with the use of colchicine.

### 3.2. Circulating Biomarkers and Imaging to Assess Residual Inflammatory Risk 

Although the role of inflammation as a critical contributor to atherosclerosis has been acknowledged for several years, a systematic assessment of the RIR to identify patients at enhanced cardiovascular risk is not implemented in clinical practice. Among the circulating inflammatory indexes, hsCRP is the biomarker most investigated to quantify RIR and establish the potential benefit of anti-inflammatory interventions in the cardiovascular field. It has been consistently found that hsCRP levels <2 mg/dL are associated with a lower risk of cardiovascular events [41,42,43].

Beyond hsCRP, additional biomarkers have been investigated as potential indicators of systemic inflammation and RIR, including IL-6, IL-1b, IL-1 receptor antagonist, or lipoprotein-associated phospholipase A2 [58]. Unfortunately, these circulating biomarkers have been demonstrated to have low specificity for vascular inflammation and seem to have only modest predictive power, overestimating the risk in primary prevention [59].

To compensate for the relative lack of specificity of such circulating biomarkers, some imaging techniques have been evaluated to assess the RIR in primary and secondary CVD prevention (Figure 3). 

Coronary computed tomography angiography (CCTA) is an increasingly available technique that can identify obstructive and/or high-risk plaque and spotty calcification and define the global coronary plaque burden [60]. Although CCTA cannot measure vessel inflammation, the detection of some features typical of a vulnerable plaque may provide incremental predictive value for future coronary events. An association between hsCRP levels and atherosclerotic lesions with high-risk features identified by CCTA has been observed [61]. Unfortunately, the subjective assessment of some plaque features as well as the increase in vascular calcification induced by statins make the role of CCTA in secondary prevention challenging. In general, the risk reclassification was reported to have greater power in younger patients, women, and in those with non-obstructive coronary artery disease (low-risk groups). Conversely, the evaluation of coronary inflammation by the analysis of perivascular fat is a novel and promising technique. The rationale of this novel method is based on the presence and evaluation of the adipocyte size gradient around the coronary artery wall due to the onset of lipolysis triggered by vascular inflammation. Computed tomography (CT) angiographic images able to quantify perivascular inflammation seem to be able to monitor plaque progression and detect its instability [62].

Hybrid positron emission tomography (PET)/CT and PET/magnetic resonance (MR) have also been used as non-invasive imaging modalities for the evaluation of RIR at the vascular level. Different radioactive tracers, such as 18-fluorodeoxyglucose (18F-FDG), may be taken up by metabolically-active cells (such as macrophages), allowing inflammation burden to be estimated in the arterial wall. Unfortunately, this approach may be adopted only to evaluate the aorta and/or the carotid arteries due to the increased background noise and high 18F-FDG uptake by the myocardium [63].

Sodium fluoride 18F (18F-NaF) has a higher specificity for coronary artery inflammation and can identify ulcerated coronary plaques. Gallium 68 (68Ga)-DOTATATE, a somatostatin receptor subtype-2 (SST2)-binding PET tracer, tracks M1-primed pro-inflammatory macrophages and its uptake is increased in coronary culprit lesions. Indeed, this tracer has been found to distinguish with great specificity high-risk from low-risk plaques [64]. Similarly, CXC-motif chemokine receptor 4 (CXCR4) imaging using 68Ga-pentoxifaxor tracer or PET/MR imaging systems of oxidation-specific epitopes using a zirconium-89 (89Zr)-labelled tracer (89Zr-LA25) or choline-based tracers are other promising tools to identify inflamed atherosclerotic lesions [65,66,67]. However, it is worth highlighting that PET is currently an expensive imaging technique, is not widely available, and is associated with high radiation exposure [59].

## 4. Conclusions

Experimental and clinical evidence has shown a pathophysiologic role and common biological pathways linking Lp(a) and inflammation in CVD. Indeed, the persistent residual cardiovascular risk despite optimal management of traditional atherosclerotic risk factors may be attributed to high Lp(a) levels and chronic inflammation. Of note, Lp(a) itself promotes inflammatory processes, and vice versa inflammatory conditions are associated with increased Lp(a) levels [68].

The quantification of both Lp(a) and hsCRP, the latter one of the most studied inflammatory biomarkers in the setting of CVDs, may help estimate residual cardiovascular disease risk [69]. In individuals without known cardiovascular disease, the presence of a pro-inflammatory status, as detected by high hsCRP levels, has been found to increase the risk of cardiovascular events associated with high Lp(a) levels [70]. Furthermore, in patients with established CVD, higher Lp(a) levels were associated with cardiovascular events only in individuals with hsCRP > 2 mg/L [71].

In addition, the finding of a reduction in Lp(a) levels associated with anti-inflammatory treatments, such as ziltivekimab [54], and of anti-inflammatory effects associated with Lp(a) level reduction, such as observed with powerful Lp(a)-lowering treatments [28], further supports the bidirectional relationship between Lp(a) and inflammation and the potential cardiovascular benefit of treatments that have them as target. However, further studies are needed to establish whether the potential impact of Lp(a)-lowering agents on cardiovascular risk is influenced by the presence of systemic inflammation as assessed by hsCRP measurement.

Recognizing the presence of high Lp(a) levels and inflammation may help refine a tailored risk management strategy. Currently Lp(a) levels >125 nmol/L (50 mg/dL) and hsCRP levels >2 mg/l are the most common parameters to define an increased risk associated with these risk factors. However, no standard strategies have been established to stratify individual atherosclerotic risk on the basis of these risk factors in clinical practice. Further research is needed to better define in which settings it is recommended to measure Lp(a) levels and assess inflammatory burden in order to improve risk stratification and how these measures may be used to monitor atherosclerotic disease progression and guide tailored therapeutic interventions. Research in these fields may lead to novel anti-atherosclerotic approaches.

## Figures and Tables

**Figure 1 jcm-12-02529-f001:**
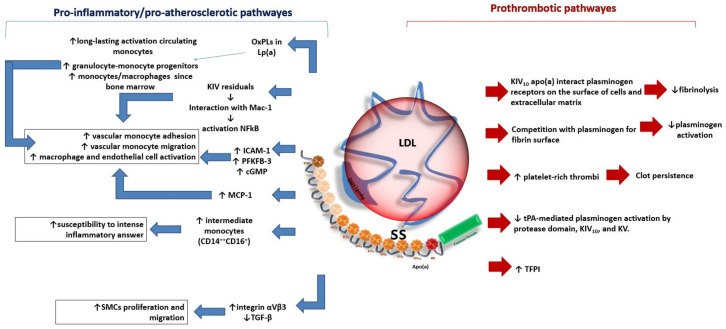
The potential impact of lipoprotein(a) (Lp(a)) on atherosclerotic and thrombotic processes. The figure represents the main hypothetic mechanisms involved in the pathogenesis of atherosclerosis and thrombosis in patients with higher Lp(a) serum concentrations. Apo(a): apolipoprotein(a); CD: cluster of differentiation; cGMP: guanosine 3′,5′-cyclic monophosphate; ICAM-1: intercellular adhesion molecule-1; KIV: kringle IV; KV: kringle V; LDL: low density lipoprotein; MCP-1: monocyte chemoattractant protein-1; NFkB: nuclear factor kB; OxPLs: oxidized phospholipids; PFKFB-3: 6-phophofructo-2-kinase/fructose-2,6-biphosphatase enzyme; SMC: smooth muscle cell; TFPI: tissue factor pathway inhibitor; TGF-β: transforming growth factor-beta; tPA: tissue plasminogen activator.

**Figure 2 jcm-12-02529-f002:**
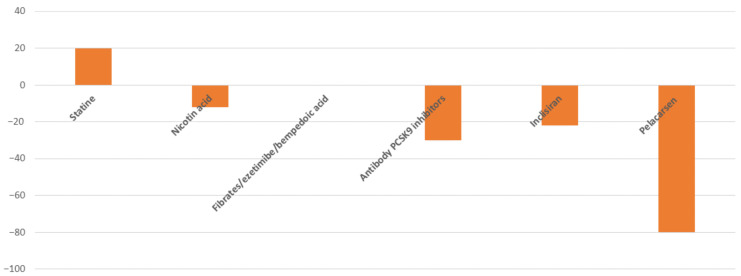
Impact of lipid-lowering treatments on Lp(a) serum concentrations.

**Figure 3 jcm-12-02529-f003:**
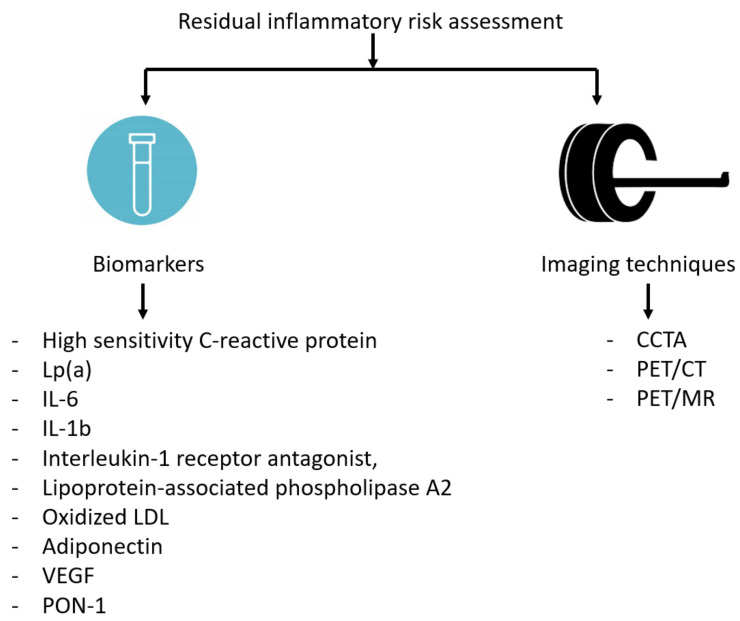
Biomarkers and imaging techniques for estimation of residual inflammatory risk in patients with cardiovascular disease. CCTA: Coronary computed tomography angiography; CT: computed tomography; Lp(a): Lipoprotein(a); IL-6 Interleukin-6; IL-1b: Interleukin-1b; LDL: Low-density lipoprotein; MR, magnetic resonance; PET, positron emission tomography; VEGF: Vascular endothelial growth factor; PON-1: Paraoxonase-1.

**Table 1 jcm-12-02529-t001:** Main clinical randomized, placebo-controlled, double-blind trials that investigated anti-inflammatory drugs to reduce atherosclerotic cardiovascular risk.

Trial	Tested Drug	Mechanism of Action	Study Design	N Patients	Study Population	Follow-Up Duration	Main Findings
CANTOS [47]	Canakinumab	IL-1β inhibition	Randomized Controlled Double-blind Phase 3 Trial	10,061	Previous MI and high CRP levels	3.7 years *	Canakinumab reduced cardiovascular events
COLCOT [50]	Colchicine	Tubulin polymerization and inflammasome inhibition	Randomized Controlled Double-blind Phase 3 Trial	4745	Patients who had had a MI within 30 days before recruitment	22.6 months *	Significant reduction in ischemic cardiovascular events
LoDoCo2 [51]	Colchicine	Tubulin polymerization and inflammasome inhibition	Randomized Controlled Double-blind Phase 3 Trial	5522	Chronic coronary disease	28.6 months *	Colchicine vs. placebo reduced cardiovascular event risk
VCUART3 [53]	Anakinra	IL-1 receptor antagonist	Randomized Controlled Double-blind Phase 2 Trial	99	ST-segment-elevation myocardial infarction	12 months	Significant reduction in hsCRP area under the curve during the first 14 days. Lower incidence of HF.
RESCUE [54]	Ziltivekimab	IL-6 inhibition	Randomized Controlled Double-blind Phase 2 Trial	264	Chronic kidney disease and high CRP levels	24 weeks	Ziltivekimab (dose dependent) reduced inflammation and thrombosis biomarkers
Tocilizumab in NSTEMI patients (ClinicalTrials.gov, NCT01491074) [55]	Tocilizumab	IL-6 receptor inhibition	Randomized Controlled Double-blind Phase 2 trial	117	NSTEMI patients scheduled for angiography	6 months	Tocilizumab reduced hsCRP levels and troponin T release after PCI
ASSAIL-MI [56]	Tocilizumab	IL-6 receptor inhibition	Randomized Controlled Double-blind Phase 2 trial	199	STEMI patients within 6 h of symptom onset	6 months	Greater myocardial salvage index (Measured with CMR after 3–6 days)
CIRT [38]	Metothrexate	Nucleotide synthesis inhibition leading to suppression of inflammation	Randomized Controlled Double-blind Phase 3 Trial	4786	Previous MI or multivessel CAD and Type 2 diabetes or metabolic syndrome	2.3 years *	Low-dose methotrexate did not result in lower IL-1B/ IL-6 or CRP levels

* Median. CAD indicates coronary artery disease; CMR, cardiac magnetic resonance; CRP, C-reactive protein; HF, heart failure; IL, interleukin; MI, myocardial infarction; NSTEMI, non-ST-segment elevation myocardial infarction; STEMI, ST-segment elevation myocardial infarction.

## Data Availability

Not applicable.

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
