# Peer review of "Lipoprotein (a), Inflammation, and Atherosclerosis"

_jcm, 2023, doi:10.3390/jcm12072529_

Round 1

Reviewer 1 Report

This comprehensive review summarizes current knowledge on the significance of Lp(a) and inflammation for the residual risk of cardiovascular diseases in appropriately treated patients. The manuscript is well-designed and essential topics are covered. However, several improvements should be made:

1. Anti-atherotic effects of Lp(a) are listed in the Figure 1, but it would be very useful if these mechanisms are briefly described and explained in the main text.

2. Section 2.2 Giving that the determination of Lp(a) is not a routine laboratory analysis, the authors should briefly mentioned analytical approach in measuring Lp(a) in biological samples, as well the most important limitations.

3. Possible combination of Lp(a) and inflammatory markers in the quantification of residual cardiovascular risk should be addressed in the final sections of the manuscript.

Author Response

Subject: revised manuscript “Lipoprotein (a), inflammation, and atherosclerosis”

Dear Editor and Reviewers,

Thank you for your response to our submission.

We are pleased to resubmit for publication the revised version of the manuscript entitled “Lipoprotein (a), inflammation, and atherosclerosis”.

All of the reviewers’ comments have been highly regarded and we have addressed each one.

Please find below our detailed responses to the reviewer's comments and the changes made to the manuscript in accordance with these.

The reviewer’s comments are underscored and our responses are below each comment. We have reported in bold the changes made in the revised manuscript and we have indicated where these changes appear by using page and line numbers.

Reviewer 1

This comprehensive review summarizes current knowledge on the significance of Lp(a) and inflammation for the residual risk of cardiovascular diseases in appropriately treated patients. The manuscript is well-designed and essential topics are covered. However, several improvements should be made:

  1. Anti-atherotic effects of Lp(a) are listed in the Figure 1, but it would be very useful if these mechanisms are briefly described and explained in the main text.

Thank you for this remark. As suggested, in the revised manuscript we have briefly described the main mechanisms underlying the atherogenic effects of Lp(a):

Lp(a) particles can cross the endothelial barrier, be retained in the arterial wall, and promote atherosclerotic plaque growth. Oxidized phospholipids carried by Lp(a) trigger macrophage apoptosis and may promote atherosclerotic lesion transformation into “vulnerable” plaques.

Lp(a) seems to contribute to arterial vessel wall inflammation by promoting monocyte cell extravasation and endothelial cell activation. Experimental studies have shown that these effects may be ascribed to adhesion molecule, e.g., intercellular adhesion molecule-1 (ICAM-1), transcription and translation upregulation and to increased activity of the enzyme 6-phophofructo-2-kinase/fructose-2,6-biphosphatase (PFKFB)-3 induced by Lp(a).

In addition, apo(a) KIV domains seem to be involved in the interaction with beta2-integrin Mac-1, which induces nuclear factor kB (NFkB) activation and leads to the production of molecules that mediate the adhesion of monocytes to the endothelium and subsequent arterial wall invasion. In vitro studies have also found that apo(a) is able to stimulate vascular smooth muscle cell proliferation and migration. The apo(a) KIV10 domain seems to interact with plasminogen receptors on the cell surface and in the extracellular matrix, thus competing in fibrinolytic processes. The binding of Lp(a) to fibrin prevents plasminogen activation and results in impaired clot degradation. Furthermore, Lp(a) has been found to be able to bind and inactivate tissue factor pathway inhibitors. Of note, most of the mechanisms that explain potential Lp(a) prothrombotic effects have been found in in vitro studies and their impact on atherothrombotic events must be confirmed in clinical settings. (Page 2, lines 20-37; page 3, lines 1-3)

  1. Section 2.2 Giving that the determination of Lp(a) is not a routine laboratory analysis, the authors should briefly mentioned analytical approach in measuring Lp(a) in biological samples, as well the most important limitations.

We acknowledge the relevance of the point raised by the reviewer. In the revised manuscript we report the currently available analytical methods for Lp(a) measurement and their limitations as follows:

Two main challenges limit Lp(a) plasmatic level assessment in clinical practice: the absence of standardized measurement methods and established target levels. (Page 4, lines 52-53)

At present, no specific method for Lp(a) concentration measurement has been established as a gold standard. Interindividual apo(a) size heterogeneity poses some difficulties in Lp(a) concentration quantification. Indeed, most of the available analytical methods for Lp(a) measurement use immunoassays that are apo(a)-isoform sensitive, and therefore may overestimate or underestimate Lp(a) levels based on apo(a) size. The Marcovina assay, which uses monoclonal antibodies, is the least apo(a)-size-sensitive immunoassay currently available on the market. Although some available assays measure Lp(a) in mass and others in molar units, in clinical practice it is recommended to measure Lp(a) plasmatic concentrations in molar units, as this measurement quantifies Lp(a) particles irrespective of the particle molecular mass. Furthermore, conversion between the two units is not advisable. Recently, liquid chromatography tandem mass spectrometry assay, an isoform-independent assay, has been validated and proposed as a reference method. (Page 5, lines 15-28)

  1. Possible combination of Lp(a) and inflammatory markers in the quantification of residual cardiovascular risk should be addressed in the final sections of the manuscript.

In accordance with this reviewer’s comment, we have specified:

The quantification of both Lp(a) and hsCRP, the latter one of the most studied inflammatory biomarkers in the setting of CVDs, may help estimate residual cardiovascular disease risk [69]. In individuals without known cardiovascular disease, the presence of a pro-inflammatory status, as detected by high hsCRP levels, has been found to increase the risk of cardiovascular events associated with high Lp(a) levels [70]. Furthermore, in patients with established CVD, higher Lp(a) levels were associated with cardiovascular events only in individuals with hsCRP > 2 mg/L [71]. (Page 10, lines 19-25)

We thank the reviewer again for taking the time to review this paper.

We look forward to hearing from you.

Sincerely,

Dr. Di Fusco Stefania Angela

Clinical and Rehabilitation Cardiology Unit, Director Prof. Furio Colivicchi

San Filippo Neri Hospital

Rome

Reviewer 2 Report

The paper presents an important and current problem concerning the importance of lp(a) and inflammation  in development of cardiovascular diseases however, due to the wide access to the literature in this area, in order  to to increase scientific soundness  and originality some corrections should be made:

Authors pay too little attention to the potential possible mechanisms of the atherosclerotic effect of lp(a).  

Too little attention is also  devoted to therapies with which the greatest hope is currently associated . 

In chapter How to measure and quantify the risk associated with high Lp(a) levels we do not find any information  for example about diagnostic problems related to the high nature of lipoproteins.

Citation quality also needs to be refined For example on page: 6 lines: 234-326 the authors mention the potential benefits of recombinant antibody against interleukin-1 receptor however they do not provide any literature references.

My  doubts are also raised by the connection lp(a) and inflammation in one  review, or rather its justification.

Author Response

Subject: revised manuscript “Lipoprotein (a), inflammation, and atherosclerosis”

Dear Editor and Reviewers,

Thank you for your response to our submission.

We are pleased to resubmit for publication the revised version of the manuscript entitled “Lipoprotein (a), inflammation, and atherosclerosis”.

All of the reviewers’ comments have been highly regarded and we have addressed each one.

Please find below our detailed responses to the reviewer's comments and the changes made to the manuscript in accordance with these.

The reviewer’s comments are underscored and our responses are below each comment. We have reported in bold the changes made in the revised manuscript and we have indicated where these changes appear by using page and line numbers.

Reviewer 2

The paper presents an important and current problem concerning the importance of lp(a) and inflammation  in development of cardiovascular diseases however, due to the wide access to the literature in this area, in order  to to increase scientific soundness  and originality some corrections should be made:

Authors pay too little attention to the potential possible mechanisms of the atherosclerotic effect of lp(a).

We thank the reviewer for this comment and accordingly we have briefly described potential mechanisms underlying the pro-atherogenic effects of Lp(a).

Lp(a) particles can cross the endothelial barrier, be retained in the arterial wall, and promote atherosclerotic plaque growth. Oxidized phospholipids carried by Lp(a) trigger macrophage apoptosis and may promote atherosclerotic lesion transformation into “vulnerable” plaques.

Lp(a) seems to contribute to arterial vessel wall inflammation by promoting monocyte cell extravasation and endothelial cell activation. Experimental studies have shown that these effects may be ascribed to adhesion molecule, e.g., intercellular adhesion molecule-1 (ICAM-1), transcription and translation upregulation and to increased activity of the enzyme 6-phophofructo-2-kinase/fructose-2,6-biphosphatase (PFKFB)-3 induced by Lp(a).

In addition, apo(a) KIV domains seem to be involved in the interaction with beta2-integrin Mac-1, which induces nuclear factor kB (NFkB) activation and leads to the production of molecules that mediate the adhesion of monocytes to the endothelium and subsequent arterial wall invasion. In vitro studies have also found that apo(a) is able to stimulate vascular smooth muscle cell proliferation and migration. The apo(a) KIV10 domain seems to interact with plasminogen receptors on the cell surface and in the extracellular matrix, thus competing in fibrinolytic processes. The binding of Lp(a) to fibrin prevents plasminogen activation and results in impaired clot degradation. Furthermore, Lp(a) has been found to be able to bind and inactivate tissue factor pathway inhibitors. Of note, most of the mechanisms that explain potential Lp(a) prothrombotic effects have been found in in vitro studies and their impact on atherothrombotic events must be confirmed in clinical settings. (Page 2, lines 20-37; page 3, lines 1-3)

Too little attention is also devoted to therapies with which the greatest hope is currently associated.

We agree with this remark and in the revised manuscript we include more information about novel Lp(a) lowering treatments.

Olpasiran, a further siRNA targeting apo(a) mRNA, has been found to result in an up to 100% placebo-adjusted mean percentage reduction in Lp(a) with a 225-mg dose administered every 24 weeks [26]. Pelacarsen, an antisense oligonucleotide (ASO) that inhibits apo(a) mRNA translation in hepatocytes, decreases Lp(a) by 36-80% based on drug dose [27]. Pe-lacarsen – an antisense oligonucleotide (ASO) which also inhibits the translation of apoprotein(a) mRNA in the hepatocyte –decreases Lp(a) by 36 - 80% based on drug dose [20]. Specifically, in patients with established cardiovascular disease, pelacarsen reduced Lp(a) by 80% when administered at a weekly dose of 20 mg, with a persistent effect at 113 days from last administration [27]. Furthermore, in patients with high Lp(a) plasmatic concentrations, pelacarsen has been found to lead to lower inflammatory activity and transendothelial migration of circulating monocytes [28]. Further studies are necessary to establish the clinical impact of these effects, which were not found with powerful LDL-C lowering agents. The main adverse effects of pelacarsen seem to be limited to injection-site reactions, which are generally mild. The ongoing Lp(a)-HORIZON study (NCT04023552), which enrolled 8324 patients with established cardiovascular disease, will provide data on the effect of 80 mg of monthly subcutaneous injection of pelacarsen on cardiovascular outcomes. (Page 4, lines 15-32)

-In chapter How to measure and quantify the risk associated with high Lp(a) levels we do not find any information for example about diagnostic problems related to the high nature of lipoproteins.

We thank the reviewer for this comment. In the revised manuscript, we have pointed out the challenges in Lp(a) plasmatic concentration measurement due to interindividual apolipoprotein(a) size heterogeneity as follows:

Two main challenges limit Lp(a) plasmatic level assessment in clinical practice: the absence of standardized measurement methods and established target levels. (Page 4, lines 52-53)

At present, no specific method for Lp(a) concentration measurement has been established as a gold standard. Interindividual apo(a) size heterogeneity poses some difficulties in Lp(a) concentration quantification. Indeed, most of the available analytical methods for Lp(a) measurement use immunoassays that are apo(a)-isoform sensitive, and therefore may overestimate or underestimate Lp(a) levels based on apo(a) size. The Marcovina assay, which uses monoclonal antibodies, is the least apo(a)-size-sensitive immunoassay currently available on the market. Although some available assays measure Lp(a) in mass and others in molar units, in clinical practice it is recommended to measure Lp(a) plasmatic concentrations in molar units, as this measurement quantifies Lp(a) particles irrespective of particle molecular mass. Furthermore, conversion between the two units of measurement is not advisable. Recently, liquid chromatography tandem mass spectrometry assay, an isoform-independent assay, has been validated and proposed as a reference method. (Page 5, lines 15-28)

-Citation quality also needs to be refined For example on page: 6 lines: 234-326 the authors mention the potential benefits of recombinant antibody against interleukin-1 receptor however they do not provide any literature references.

Thank you for pointing out this aspect. In the revised manuscript, we have added some literature references to the potential benefits of recombinant interleukin-1 antagonists as follows:

Clinical studies on patients with ACS showed a reduction in hsCRP plasmatic concentrations without a clear benefit on ischemic risk [Abbate A, Kontos MC, Abouzaki NA, Melchior RD, Thomas C, Van Tassell BW, Oddi C, Carbone S, Trankle CR, Roberts CS, Mueller GH, Gambill ML, Christopher S, Markley R, Vetrovec GW, Dinarello CA, Biondi-Zoccai G. Comparative safety of interleukin-1 blockade with anakinra in patients with ST-segment elevation acute myocardial infarction (from the VCU-ART and VCU-ART2 pilot studies). Am J Cardiol. 2015 Feb 1;115(3):288-92]. However, treatment with anakinra, a recombinant human IL-1 receptor antagonist, has been found to be associated with a reduced risk of death and heart failure in patients with ST‐segment-elevation myocardial infarction [Abbate A, Trankle CR, Buckley LF, Lipinski MJ, Appleton D, Kadariya D, Canada JM, Carbone S, Roberts CS, Abouzaki N, Melchior R, Christopher S, Turlington J, Mueller G, Garnett J, Thomas C, Markley R, Wohlford GF, Puckett L, Medina de Chazal H, Chiabrando JG, Bressi E, Del Buono MG, Schatz A, Vo C, Dixon DL, Biondi-Zoccai GG, Kontos MC, Van Tassell BW. Interleukin-1 Blockade In-hibits the Acute Inflammatory Response in Patients With ST-Segment-Elevation Myo-cardial Infarction. J Am Heart Assoc. 2020 Mar 3;9(5):e014941. doi: 10.1161/JAHA.119.014941]. (Page 7, lines 5-9)

We also reported this last study in Table 1, which summarizes the main randomized, placebo-controlled, double-blind, clinical trials that investigated anti-inflammatory drugs to reduce atherosclerotic cardiovascular risk.

-My doubts are also raised by the connection lp(a) and inflammation in one review, or rather its justification.

We thank the reviewer for this comment. In the introduction we specify:

Furthermore, Lp(a) pathophysiologic effects and inflammatory processes share common biological pathways that contribute to atherogenesis. (Page 1, lines 55-57)

In the conclusion section of the revised manuscript, we also have pointed out the link between Lp(a) and inflammation.

Experimental and clinical evidence has shown a pathophysiologic role and common biological pathways linking Lp(a) and inflammation in CVD. (Page 10, lines 13-14)

Of note, Lp(a) itself promotes inflammatory processes, and vice versa inflammatory conditions are associated with increased Lp(a) levels. (Page 10, lines 16-18)

In individuals without known cardiovascular disease, the presence of a pro-inflammatory status as detected by high hsCRP levels has been found to increase the risk of cardiovascular events associated with high Lp(a) levels. Furthermore, in patients with established cardiovascular disease, higher Lp(a) levels were associated with cardiovascular events only in individuals with hsCRP > 2 mg/L. In addition, the finding of a reduction in Lp(a) levels associated with anti-inflammatory treatments, such as ziltivekimab [54], and of anti-inflammatory effects associated with Lp(a) level reduction, such as observed with powerful Lp(a)-lowering treatments [28], further supports the bidirectional relationship be-tween Lp(a) and inflammation and the potential cardiovascular benefit of treatments that have them as target. However,  further studies are needed to establish whether the potential impact of Lp(a)-lowering agents on cardiovascular risk is influenced by the presence of systemic inflammation as assessed by hsCRP measurement. (Page 10, lines 21-33)

We thank the reviewer again for taking the time to review this paper.

We look forward to hearing from you.

Sincerely,

Dr. Di Fusco Stefania Angela

Clinical and Rehabilitation Cardiology Unit, Director Prof. Furio Colivicchi

San Filippo Neri Hospital

Rome

Round 2

Reviewer 2 Report

Thank you for taking into account all comments and making the appropriate changes.

I'm just wondering if writing about the measurement of plasma lp(a) concentration is accurate since most routine determinations measure the of serum lp(a) concentration.

Author Response

Dear Editor and Reviewers,

Thank you for your response to our submission.

We are pleased to resubmit for publication the revised version of the manuscript entitled “Lipoprotein (a), inflammation, and atherosclerosis”.

Please find below our responses to the reviewer's comment.

Reviewer 2

Thank you for taking into account all comments and making the appropriate changes.

I'm just wondering if writing about the measurement of plasma lp(a) concentration is accurate since most routine determinations measure the of serum lp(a) concentration.

Due to reviewer concern throughout all the revised manuscript, we avoid the use of plasmatic Lp(a) concentration and where necessary we substituted the term “plasma” with “serum” concentration. (page 2 line 65; page 3 line 90; page 4 line148; page 5 line 200)

Best regards,

Dr. Di Fusco Stefania Angela

Clinical and Rehabilitation Cardiology Unit, Director Prof. Furio Colivicchi

San Filippo Neri Hospital

Rome